# Humans treat unreliable filled-in percepts as more real than veridical ones

Benedikt V Ehinger[1]*, Katja Häusser[1], José P Ossandón[1,2†], Peter König[1,3†]

[1]Neurobiopsychology, Institute of Cognitive Science, University of Osnabrück, Osnabrück, Germany; [2]Biological Psychology and Neuropsychology, University of Hamburg, Hamburg, Germany; [3]Department of Neurophysiology and Pathophysiology, University Medical Center Hamburg-Eppendorf, Hamburg, Germany

**Abstract** Humans often evaluate sensory signals according to their reliability for optimal decision-making. However, how do we evaluate percepts generated in the absence of direct input that are, therefore, completely unreliable? Here, we utilize the phenomenon of filling-in occurring at the physiological blind-spots to compare partially inferred and veridical percepts. Subjects chose between stimuli that elicit filling-in, and perceptually equivalent ones presented outside the blind-spots, looking for a Gabor stimulus without a small orthogonal inset. In ambiguous conditions, when the stimuli were physically identical and the inset was absent in both, subjects behaved opposite to optimal, preferring the blind-spot stimulus as the better example of a collinear stimulus, even though no relevant veridical information was available. Thus, a percept that is partially inferred is paradoxically considered more reliable than a percept based on external input. In other words: Humans treat filled-in inferred percepts as more real than veridical ones.

*For correspondence: behinger@uos.de

†These authors contributed equally to this work

Competing interests: The authors declare that no competing interests exist.

## Introduction

In order to make optimal and adaptive decisions, animals integrate multiple sources of sensory information across time and space. One of the prime examples of this is observed when animals are confronted with coherently-moving stimuli during random-dot motion experiments. In such experiments, performance and the corresponding neural activity vary proportionally to signal strength in a way that is consistent with the progressive integration of evidence over time (*Shadlen et al., 1996*; *Shadlen and Newsome, 2001*). Besides temporal accumulation, sensory integration is also possible by combining the information from multiple sensory sources (*Quigley et al., 2008*; *Schall et al., 2009*; *Hollensteiner et al., 2015*; *Wahn and König, 2015a*, *2015b*, *2016*).

In the case of multisensory perception, several experiments have shown that integration often occurs in a statistically optimal way. This has been best demonstrated in cue-integration tasks in which humans perform as if they were weighting the different sources of information according to their respective reliabilities (*Ernst and Banks, 2002*; *Alais and Burr, 2004*; *Körding and Wolpert, 2004*; *Tickle et al., 2016*). This form of statistical inference has also been demonstrated for cortical neurons of the monkey brain, with patterns of activity at the population level that are consistent with the implementation of a probabilistic population code (*Gu et al., 2008*; *Fetsch et al., 2011*).

In most of these sensory integration experiments, the perceptual reliability of different inputs is probed through quantitative manipulations of the inputs' signal-to-noise ratios (*Heekeren et al., 2004*; *Tassinari et al., 2006*; *Bankó et al., 2011*). However, some percepts are unreliable not because they are corrupted by noise but because they are inferred only from the context and thus intrinsically uncertain. This occurs naturally in monocular vision at the physiological blind spot, where content is 'filled-in' based on information from the surroundings. In this case, no veridical percept is

**eLife digest** To make sense of the world around us, we must combine information from multiple sources while taking into account how reliable they are. When crossing the street, for example, we usually rely more on input from our eyes than our ears. However, we can reassess the reliability of the information: on a foggy day with poor visibility, we might prioritize listening for traffic instead.

But how do we assess the reliability of information generated within the brain itself? We are able to see because the brain constructs an image based on the patterns of activity of light-sensitive proteins in a part of the eye called the retina. However, there is a point on the retina where the presence of the optic nerve leaves no space for light-sensitive receptors. This means there is a corresponding point in our visual field where the brain receives no visual input from the outside world. To prevent us from perceiving this gap, known as the visual blind spot, the brain fills in the blank space based on the contents of the surrounding areas. While this is usually accurate enough, it means that our perception in the blind spot is objectively unreliable.

To find out whether we are aware of the unreliable nature of stimuli in the blind spot, Ehinger et al. presented volunteers with two striped stimuli, one on each side of the screen. The center of some of the stimuli were covered by a patch that broke up the stripes. The volunteers' task was to select the stimulus with uninterrupted stripes. The key to the experiment is that if the central patch appears in the blind spot, the brain will fill in the stripes so that they appear to be continuous. This means that the volunteers will have to choose between two stimuli that both appear to have continuous stripes. If they have no awareness of their blind spot, we might expect them to simply guess. Alternatively, if they are subconsciously aware that the stimulus in the blind spot is unreliable, they should choose the other one.

In reality, exactly the opposite happened: the volunteers chose the blind spot stimulus more often than not. This suggests that information generated by the brain itself is sometimes treated as more reliable than sensory information from the outside world. Future experiments should examine whether the tendency to favor information generated within the brain over external sensory inputs is unique to the visual blind spot, or whether it also occurs elsewhere.

possible at the blind spot location. Even though changes in reliability due to noise directly result in behavioral consequences, the effects of the qualitative difference between veridical and inferred percepts, that are otherwise apparently identical, are unknown.

We recently reported differences in the processing of veridical and inferred information at the level of EEG responses (*Ehinger et al., 2015*). We demonstrated that a qualitative assessment of differences in reliability exists at the neural level in the form of low- and high-level trans-saccadic predictions of visual content. Notably, active predictions of visual content differed between inferred and veridical visual information presented inside or outside the blind spot. Although no difference was found between low-level error signals, high-level error signals differed markedly between predictions based on inferred or veridical information. We concluded that the inferred content is processed *as if* it were veridical for the visual system, but knowledge of its reduced precision is nevertheless preserved for later processing stages.

In the present experiment, we address whether such an assessment of a dichotomous, qualitative difference in reliability is available for perceptual decision-making. Using 3D shutter glasses, we presented one stimulus partially in the participant's blind spot to elicit filling-in and a second stimulus at the same eccentricity in the nasal field of view outside of the blind spot. The subject's task was to indicate which of the two stimuli was continuously striped and did not present a small orthogonal inset (see *Figure 1a*). Crucially, stimuli within the blind spot are filled-in and thus perceived as continuous, even when they present an inset. In the diagnostic trials, both stimuli were physically identical and continuous, and subjects were confronted with an ambiguous decision between veridical and partially inferred stimuli.

We evaluated two mutually exclusive hypotheses on how perceptual decision-making could proceed when confronted with an ambiguous decision between veridical and inferred percepts. In the first case, we hypothesized that subjects are unable to make perceptual decisions based on an

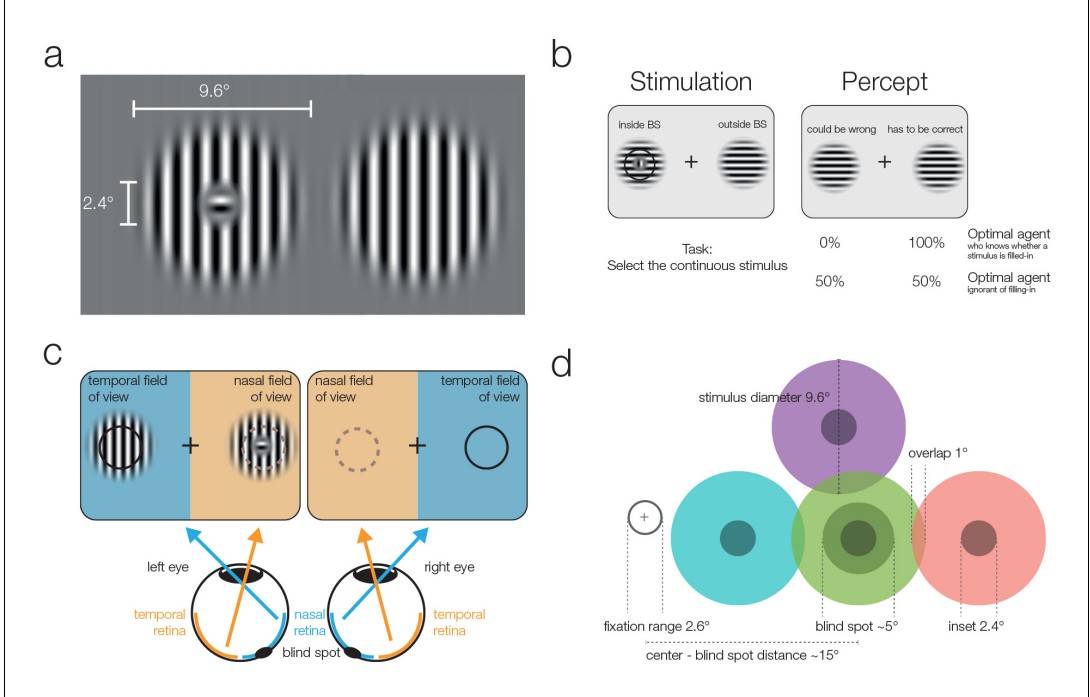

**Figure 1.** Stimuli and stimulation. (**a**) Striped stimuli used in the study. The inset was set to ~50% of the average blind spot size. The global orientation of both stimuli was the same, but in different trials it could be either vertical (as shown here) or horizontal (not shown). (**b**) Each stimulus was displayed individually either (partially) inside or (completely) outside the blind spot. This example presents an inset stimulus inside the subject's left blind spot. However, due to filling-in, it is perceived as continuous (right column). The task required subjects to select the continuous stimulus, and it was designed to differentiate between two mutually exclusive predictions: First, subjects cannot differentiate between the two different types of stimuli and thus answer randomly. Alternatively, subjects have implicit or explicit knowledge about the difference between inferred (filled-in) and veridical contents and consequently select the stimulus outside the blind spot in ambiguous trials. (**c**) Two stimuli were displayed using shutter glasses. Each stimulus was presented to one eye only, and it is possible that both are presented to the same eye (as in the example depicted here). That is, the left stimulus could be shown either in the temporal field of view (nasal retina) of the left eye (as in the plot) or in the nasal field of view (temporal retina) of the right eye (not shown). In this case, the trial was unambiguous: The stimulus with an inset was presented outside the blind spot and could be veridically observed, therefore, the correct answer was to select the left stimulus. (**d**) The locations of stimulus presentation in the five experiments. All stimuli were presented relative to the blind spot location of each subject. All five experiments included the blind spot location (green). In the second and fifth experiment, effects at the blind spot were contrasted with a location above it (purple). In the third experiment, the contrasts were in positions located to the left or the right of the blind spot. Note that both stimuli were always presented at symmetrical positions in a given trial, the position of the stimuli differed only across trials.

The following figure supplement is available for figure 1:

**Figure supplement 1.** Trial balancing of all experiments.

assessment of differences in reliability between veridical and inferred stimuli. Therefore, subjects would have an equal chance of selecting stimuli presented inside or outside the blind spot. Alternatively, it might be possible to use the information about the reduced reliability of filled-in information. In this case, we expect subjects to follow an optimal strategy and trust a stimulus presented outside the blind spot, where the complete stimulus is seen, more often than when the stimulus is presented inside the blind spot, where it is impossible to know the actual content within the filled-in part.

## Results

We conducted five experiments (see *Figure 1* and the methods for a detailed description of the tasks). The first one tested the presence of a bias against the blind spot location; the other four experiments were replications of the first experiment with additional control conditions. In the first two controls, we tested the existence of biases between the nasal and temporal visual fields at

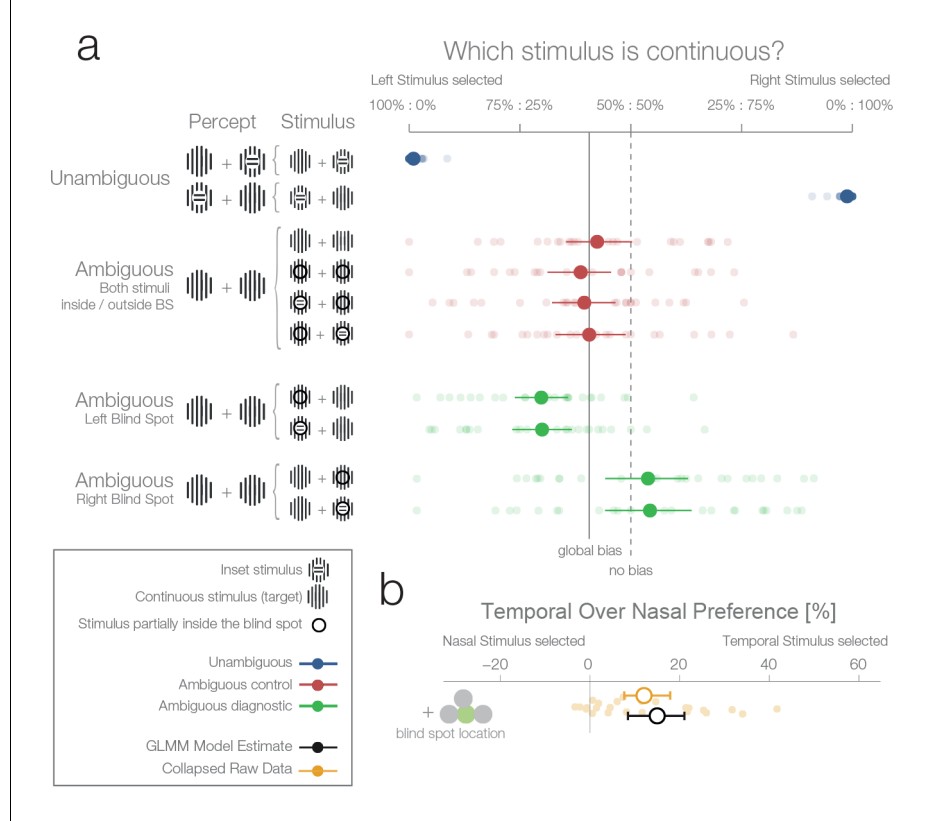

**Figure 2.** First experiment. (**a**) The left column shows schematics of the actual stimulation and the associated percepts for the corresponding data presented in the right panel. A dark-lined circle, where present, indicates that the stimulus was presented in the blind spot and, consequently, an inset stimulus within was perceived as a continuous stimulus due to filling-in. The plot to the right shows each subject's (n = 24) average response and the group average (95% bootstrapped confidence intervals, used only for visualization). The results from unambiguous trials (blue) show that subjects were almost perfect in their selection of the continuous stimulus when an inset was visible. For the first type of ambiguous control trials (red), both stimuli were presented either outside or inside the blind spot. Here, only a global bias toward the left stimulus can be observed (solid line, the mean across all observed conditions in red). Note that the performance when presenting an inset in the blind spot was identical to the one of presenting a continuous stimulus in the blind spot. The ambiguous diagnostic conditions (green) show the, unexpected, bias toward the blind spot (for either side). (**b**) Statistical differences were evaluated by fitting a Bayesian generalized mixed linear model. In the model, the left and right ambiguous diagnostic conditions were combined in a single estimate of the bias for nasal or temporal stimuli (outside or inside the blind spot respectively). The plot shows the average effect of each subject (small yellow dots), the bootstrapped summary statistics of the data (yellow errorbar), and the posterior 95% credibility interval model estimate (black errorbar).

locations other than the blind spot. In the third control, we tested whether an opposite bias existed when the task was reversed. The last experiment controls whether the observed bias could be explained by probability matching.

## Experiment 1

In the first experiment, 24 subjects performed a 2-AFC task in which they had to indicate which of two stimuli was continuously striped instead of presenting a small orthogonal central inset (**Figure 1a,b**). The stimuli were presented simultaneously in the periphery at the locations of the blind spots or at equivalent eccentricity on the opposite side (**Figure 1c,d**). We used a 3D monitor and shutter glasses that allowed for the controlled monocular display of the stimuli. That means each stimulus was visible to a single eye only. There were always two stimuli, therefore, in a given trial either one or both eyes were stimulated (**Figure 1b**). Importantly, subjects always perceived the two stimuli at the same locations, to the left and the right of the fixation cross. In this experiment

there were perceptually ambiguous trials, where two continuous stimuli were perceived, and unambiguous trials where one stimulus contained a visible inset.

In the unambiguous trials, an orthogonal inset was present in one of the stimuli. Importantly, in these trials, the stimulus with the inset was outside the blind spot and therefore clearly visible. As expected, subjects performed with near-perfect accuracy (*Figure 2*, unambiguous trials, blue data), choosing the continuous stimulus in an average of 98.8% of trials (95%-quantile over subjects [96.4–100%]).

There were two types of ambiguous trials. In the first type (*Figure 2*, ambiguous control, red data), one of the following applied: both stimuli were continuous and appeared outside the blind spots in the nasal visual fields (*Figure 2*, row 3); both were continuous and appeared inside the blind spots (*Figure 2*, row 4); or one was continuous, the other had an inset, and both appeared inside the blind spots with the inset either in the left or the right blind spot (*Figure 2*, rows 5 and 6). In the case when a stimulus with an inset was present, this central part was perfectly centered inside the blind spot (*Figure 1a*), and in consequence was perceived as continuous due to filling-in. Thus, in all four versions, subjects perceived two identical stimuli, and there was no single correct answer. In this type of ambiguous trial, subjects showed a small global leftward bias and chose the left stimulus in 53.6% of trials (*Figure 2*, continuous vertical line). In addition, no difference can be seen between the perception of pairs of filled-in stimuli and pairs of veridical continuous stimuli (*Figure 2*, rows 3 vs. 4–6). This type of ambiguous control trial confirms that filling-in was accurately controlled in our experiment.

In the second type of ambiguous trials one stimulus was presented inside and the other outside the blind spot (*Figure 2*, ambiguous diagnostic, data in green). This allowed us to test directly between two rival predictions: whether subjects will show a bias against the stimulus that is partially inferred (inset area inside the blind spot) and in favor of the veridical stimulus (in the opposite visual field), or no bias. Selecting the filled-in stimulus is a suboptimal decision because the stimulus presented partially in the blind spot is the only one which could possibly contain the inset. This is explicit in the cases where an inset is shown in the blind spot but rendered invisible by filling-in (*Figure 2a*, ambiguous trials with an inset stimulus). To analyse the data, we modeled the probability increase of choosing the right stimulus if the right stimulus was presented in either the temporal visual field of the right eye (blind spot) or the nasal visual field of the left eye (non-blind spot). A similar factor was used for the left stimulus. Subsequently, the two one-sided model estimates were collapsed to a single measure of preference for stimulus presented at the nasal or temporal visual field (outside or inside the blind spot respectively). As a model for inference, we used a Bayesian generalized mixed linear model. There were three additional factors in the model (handedness, dominant eye, and precedent answer) that are not the focus of the experiment and are thus reported in the methods section (see 'Effects not reported in the Results section').

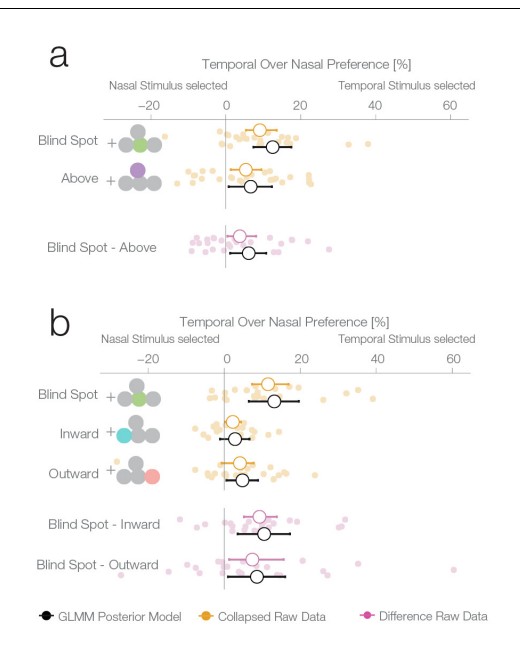

**Figure 3.** Location control experiments. Two control experiments were designed to test whether the observed bias for the blind spot could be explained by a general bias for stimuli presented in the temporal visual field. (a) Results of experiment 2. In a given trial, stimuli were presented either at the locations corresponding to the blind spot or at locations above it. Results are presented as in *Figure 2b*, with the addition of within-subject differences between blind spot and control locations (in purple). (b) Results of experiment 3. In a given trial, stimuli were presented at the locations corresponding to the blind spot or at locations to inward (toward the fixation cross) or outward (away from the fixation cross) to it. Note that the blind spot effect is replicated in both experiments. In addition, both blind spot effects are larger than in any control location.

*Figure 2a* (ambiguous diagnostic, data in green) and 2b show that subjects indeed presented a bias. However, in contrast to our expectations, subjects were more likely to choose the filled-in percept with a 15.01% preference for stimuli presented in the temporal visual field ($CDI_{95}$ 8.49–21.08%). In other words, when subjects had to decide which of the two stimuli (both perceived as being continuous, and in most cases actually physically identical) was less likely to contain an inset, they showed a bias for the one in which the critical information was not sensed directly but inferred from the surrounding spatial context. Remarkably, this result is at odds with both of our experimental predictions that postulated either no bias or a bias in favor of the veridical stimulus.

## Experiment 2

The second experiment was designed to replicate the unexpected result of the first experiment and evaluate whether the blind spot bias observed was due to systematic differences between nasal and temporal retinae. In experiment 1, we presented stimuli at mirror eccentricities inside and outside the blind spot, that is, temporal and nasal respectively (see *Figure 1c*). In experiment 2, we tested whether the bias in experiment 1 was specific to the blind spot location or related to known differences between the temporal and nasal retina (*Fahle and Schmid, 1988*). There is higher photoreceptor density (*Curcio et al., 1990*), spatial resolution (*Rovamo et al., 1982*), luminance discrimination (*Pöppel et al., 1973*) and orientation discrimination (*Paradiso and Carney, 1988*) at locations that project to the nasal retina (the temporal visual field where the blind spots are located). Thus, we repeated our experiment with a new group of subjects (n = 27) and an additional experimental condition. In this new condition, the two stimuli were displayed at symmetrical locations above the blind spot (25° above the horizontal meridian; see *Figure 1d*, purple location). The results of this second experiment replicate the observations of experiment 1 (*Figure 3a*): subjects showed a bias for selecting the stimulus presented inside the blind spot (12.5%, $CDI_{95}$ 7.35–17.49%). However, subjects also presented a bias in the control condition, toward the stimuli presented in the temporal visual field above the blind spot (6.63%, $CDI_{95}$ 0.77–12.3%). The bias was nevertheless stronger inside the blind spot (paired difference: 6.11%, $CDI_{95}$ 1.16–10.78%). In summary, additionally to the bias inside of the blind spot area, we observed that subjects also showed a smaller bias for stimuli presented to the nasal retina (temporal visual field).

## Experiment 3

To better delineate the distribution of bias across the temporal visual field and to clarify if the blind spot location is, in fact, special, we performed a third experiment on a new group of subjects (n = 24). Here, we compared biases in the blind spot to two other control conditions flanking the blind spot region from either the left or right (*Figure 3b*). The blind spot location again revealed the strongest effect of a bias for the temporal visual field (13.18% $CDI_{95}$ 6.47–19.64%), while the locations inwards and outwards resulted in a 2.85% and 4.8% bias, respectively ($CDI_{95}$ −1.1–6.65%; $CDI_{95}$ 0.58–8.89%). The bias of both control locations was different from the bias of the blind spot location (BS vs. inward: 10.51%, $CDI_{95}$ 3.55–17.29%; BS vs. outward: 8.61%, $CDI_{95}$ 0.98–16.04%). In this experiment, as in experiments 1 and 2, we observed a bias that is specific to the blind spot region.

## Experiment 4

The results of the three previous experiments suggest that subjects considered the filled-in percept a better exemplar of a continuous non-inset stimulus, in disregard of the physical possibility of the presence on an inset inside the blind spot. To confirm this, we performed a fourth experiment with a new group of subjects (n = 25). This experiment was identical to the first experiment, except that in this case, the subjects' task was to choose the stimulus with an inset, instead of the continuous one. In this case, if a filled-in stimulus is indeed considered a more reliable exemplar of a continuous stimulus, the non blind spot stimulus should be preferred in the diagnostic trials. This was the case; subjects showed a bias for selecting the stimulus presented outside the blind spot (7.74%, $CDI_{95}$ 1.56–13.68%, *Figure 4a*), thus resulting in the expected reversal of the bias pattern observed in the first three experiments. This pattern is again suboptimal, since this time the filled-in stimulus is the one that could conceal the target. The result of this experiment indicates that the observed biases do

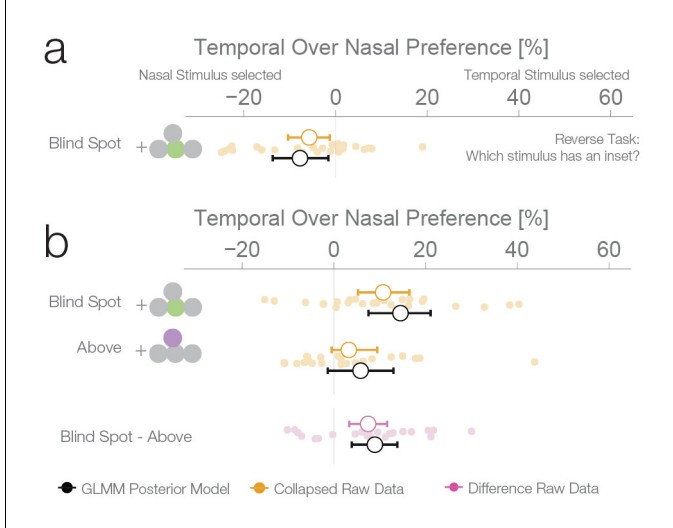

**Figure 4.** Task instruction control and probability matching control. (a) Results of experiment 4. This control was the same as experiment 1, except that subjects have to choose the stimulus with an inset (instead of the continuous one). (b) Results of experiment 5. This control was similar to experiment 2, except that no inset stimulus was ever experienced in the control location above in the temporal visual field.

The following figure supplement is available for figure 4:

**Figure supplement 1.** Correlation between experiment 4 and 5.

not correspond to an unspecific response bias for the blind spot, and instead are a consequence of considering the inferred percepts as more reliable exemplars of a continuous stimulus.

## Experiment 5

We performed a final control to evaluate whether the observed bias for a filled-in stimulus was not a result of subjects using a probability matching heuristic. It is possible that, in order to solve the ambiguous task, subjects used their knowledge of the rate of appearance of continuous and inset stimuli at different locations as learned during unambiguous trials. As it is impossible to experience an inset in the blind spot, the base rate of continuous stimuli at that location is 1.0. Therefore, when confronted with two stimuli that are apparently identical, one inside and one outside the blind spot, subjects might just apply the base rate they have learned instead of relying on a perceptual estimate. If this is the case, subjects should show a bias for the location where they experienced exclusively continuous stimuli during unambiguous trials, which could result in a bias pattern similar to the one observed in experiments 1–3. To evaluate this alternative explanation, we performed a further experiment with the same group of subjects that participated in experiment 4. Experiment 5 was similar to experiment 2, with control trials presenting stimuli above the blind spot. However, in contrast to experiment 2, subjects never experienced an inset in the temporal field in the above positions during unambiguous trials (see *Figure 1—figure supplement 1* for a detailed overview of trial randomization). This results in an identical base rate of occurrence of a continuous stimulus in the temporal field for both the above and blind spot locations. Consequently, if the behavior observed in the previous experiments was a result of probability matching, in this experiment we should observe the same bias at both the blind spot and the temporal field above locations. Subjects showed a bias for selecting the stimulus presented inside the blind spot (14.53%, $CDI_{95}$ 7.56–21.09%, *Figure 4b*), replicating again the results of experiment 1–3. At odds with the probability matching hypothesis, the bias for the temporal field in the above location was only 5.84%, not different from 0 ($CDI_{95}$ −1.33–13.01%) and similar to what was observed in experiment 2. This bias was different from the bias observed in the blind spot (paired-diff: 8.95%, $CDI_{95}$ 3.91–13.85%). The same group of subjects participated in experiment 4 and 5, allowing us to make a within subjects comparison between the two tasks. Subjects' performance in these two tasks was negatively correlated (r = −0.61,

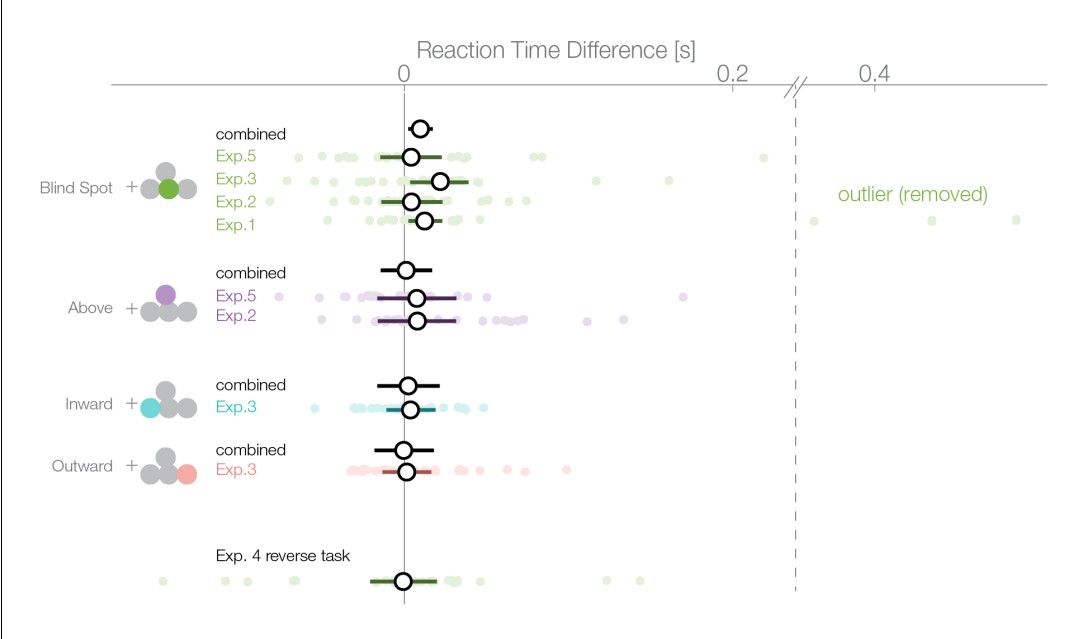

**Figure 5.** Reaction times. Reaction times of trials where the nasal stimulus was chosen minus the reaction times of trials where the temporal stimulus was chosen. Single subject estimates and 95% CI posterior effect estimates are shown. The black (combined) estimate results from a model fit of all data combined, the individual confidence intervals represent the experiment-wise model fits. We observe a reaction time effect only inside the blind spot.

p=0.002, see *Figure 4—figure supplement 1*). Taking the task reversal of experiment 4 into account, this result indicates that subjects were consistently biased to consider the inferred filled-in stimulus a better exemplar of a continuous stimulus. The result of experiment 5 thus gives evidence that the bias for the filled-in stimulus was not a consequence of subjects matching the base rate of the occurrence of different stimuli during unambiguous trials.

## Reaction time analysis

A bias for the temporal visual field, especially the blind spot, could also be reflected in the distribution of reaction times. We compared the reaction times of trials where subjects selected a stimulus in the temporal visual field against trials where the stimulus in the nasal visual field was selected. The reaction time analysis was not a planned comparisons, thus, in contrast to the other analyses presented here, it is explorative. In the first experiment, we observed an average reaction time of 637 ms (minimum subject average: 394 ms, maximum 964 ms; *Figure 5*). We used a linear mixed model to estimate the reaction time difference for selecting a stimulus presented inside the blind spot (temporally) against one outside the blind spot (nasally). In the first experiment, after excluding three outliers, we observed this effect with a median posterior effect size of 13 ms faster reaction times when selecting the blind spot region ($CDI_{95\%}$ 2–42 ms). The three outliers (on the right of the vertical dashed line in *Figure 5*) were identified visually and removed because they were distinctively different from the rest of the population. The mean of the outliers was 5.2 SD away from the remaining subjects. The outliers were nevertheless in the same direction of the reaction time effect and did not change its significance (with outliers, 63 ms, $CDI_{95}$ 7–124 ms). However, faster reaction times while selecting the blind spot stimulus were not present individually in the other four experiments. The nominal differences were in the same direction as experiment 1 but not significant (Exp.2: 4 ms, $CDI_{95}$ −14–23 ms; Exp.3: 22 ms. $CDI_{95}$ −3–39 ms; Exp.4: −1 ms $CDI_{95}$ −20–21 ms; Exp.5: 4 ms $CDI_{95}$ −15–23 ms). Non-significant results were obtained for the other locations tested (above Exp.2: 8 ms, $CDI_{95}$ −38–53 ms; above Exp.4: 8 ms $CDI_{95}$ −17–32 ms; outward: 2 ms $CDI_{95}$ −13–16 ms; inward: 4 ms, $CDI_{95}$ −29–37 ms). After combining all data (without experiment 4 as the task was

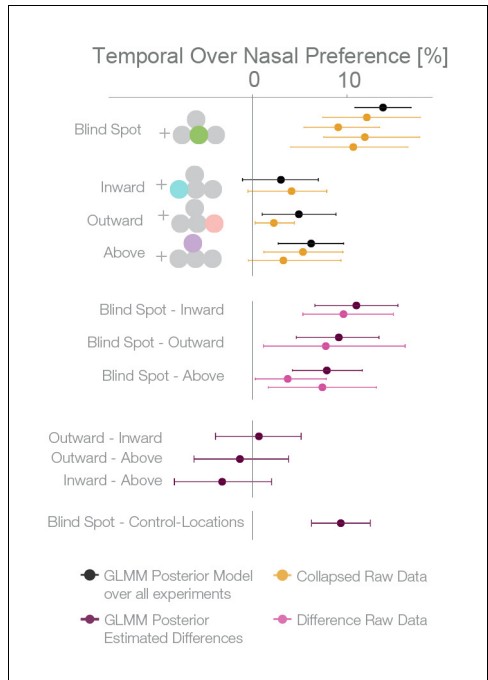

**Figure 6.** Summary and overview of blind spot effects. Posterior GLMM-effect estimates of all data combined (black) except experiment 4 (inversed task). We also show for each experiment the 95% CI of bootstrapped means summary statistics of the data (yellow). Next, we show difference values between the blind spot and all other control locations (model dark, raw data pink). As discussed in the text, the control locations outward, inward and above do not differ (fourth last to second last row), and thus we compare the blind spot effect to all locations combined (last row).

The following figure supplements are available for figure 6:

**Figure supplement 1.** Normalized data with control locations for all experiments.

**Figure supplement 2.** Posterior predictive checks.

reversed), we observed a reduced reaction time for decisions for the blind spot stimulus with 10 ms ($CDI_{95}$ 2–17 ms) but not in any other location. We do not find this small bias in any experiment individually (except Exp. 1) but only after pooling over experiments and therefore, we should interpret it cautiously. In conclusion, subjects selected stimuli in the blind spot slightly faster than stimuli outside the blind spot. The same effect does not appear for the other temporal control locations.

## Combined effect estimates over all experiments

For an overview of all experiments and the results of a Bayesian logistic mixed effects model that combines all experiments, see *Figure 6*, *Figure 6—figure supplement 1* and *Supplementary file 1*. In the combined model, we did not find any differences between the temporal field effects at locations other than the blind spots (*Figure 6*, fourth last to second last row). That is, the temporal field effects of the locations inward, outward and above were not different from each other. For the sake of clarity, we combined these location levels. Keeping everything else constant, we observed that if we present one stimulus in the blind spot against the equidistant nasal location, subjects are 13.82% ($CDI_{95}$ 10.84–16.78%, t-test, t = 8.7, df = 98, p<0.001) more likely to choose the stimulus in the blind spot. This bias is stronger than the effect observed elsewhere in the temporal field by 9.35% ($CDI_{95}$ 6.25%–12.47%; paired t-test, t = 4.8, df = 74, p<0.001). In summary, subjects showed a robust bias for the blind spot locations that could not be explained by a non-specific bias for the temporal visual field. In conclusion, when confronted with an ambiguous choice between veridical and inferred sensory information, human subjects showed a suboptimal bias for inferred information.

## Discussion

When confronted with identical physical stimulation, subjects showed a consistent bias for blind spot inferred percepts which was stronger than the bias at any other location in the temporal visual field.

Why do subjects choose the blind spot location when it is objectively the least reliable? Our interpretation takes the results at face value: subjects must possess at least implicit information about whether a percept originates from the blind spot in order to show a bias for or against it. At the same time, the veridical information from the other stimulus is also available. This indicates that perceptual decision-making can rely more on inferred than veridical information, even when there is some knowledge about the reduced reliability of the inferred input available in the brain (*Ehinger et al., 2015*). This is also supported by the results of the reaction time analyses that indicated a faster evidence accumulation for the inferred percepts. In other words, the implicit knowledge that a filled-in stimulus is objectively less reliable does not seem to be used for perceptual

decision-making. This suboptimal decision between qualitatively different veridical and inferred inputs is in contrast to properties of standard sensory integration. There, reduced reliability derived from noisy but veridical signals results in a corresponding weighting of inputs and consequently in optimal decisions (*Körding et al., 2007*). In the following, we discuss two potential explanations of this discrepancy of processing filled-in information and standard sensory integration. The first explanation focuses on physiological properties of neuronal and small circuits' response properties at and around the blind spot region. The second explanation addresses the conceptual level and uses the general notion of predictive coding.

First, although the filled-in percept is by definition independent of the stimulus within the blind spot, it is nevertheless based on the information sensed by the region around the blind spot in the nasal retina. We might assume that an area, e.g. in the nasal retina around the blind spot region, that has a lower contrast threshold also shows stronger neuronal signals for super-threshold stimuli. This could in principle lead to a filled-in stimulus with increased salience as compared to the veridical stimulus. Effectively, this explanation proposes that differences in physiological properties of nasal and temporal retinae are transferred to the filling-in process making it the 'better' candidate stimulus in an ambiguous condition. Above we already introduced some evidence for psychophysical differences between the nasal and temporal visual field (*Fahle and Schmid, 1988*). There is also some evidence for the superiority of the blind spot in a Vernier task (*Crossland and Bex, 2009*). The areas around the blind spot showed greater performance compared to areas at similar eccentric locations in the nasal visual field. It is still unclear whether this goes over and beyond the aforementioned temporal/nasal bias. Unfortunately, this explanation runs into the problem that the sensitivity in the region corresponding to the blind spot in the other eye is also enhanced compared to regions at similar eccentricities (*Wolf and Morandi, 1962*; *Midgley, 1998*). This suggests that differences between the eyes in the area around the blind spot should be the smallest within the contrast between temporal and nasal retina. Moreover, we explicitly controlled for temporal-nasal differences in experiments 2 and 3, and found that it is not enough to explain the effect specific to the blind spot. Thus, an explanation of the observed effects based on known differences in retinal properties is currently tentative at best.

An alternative explanation is based on the framework of predictive coding (*Friston et al., 2006*, *2012*; *Summerfield and de Lange, 2014*). Specifically, context information of static stimuli would be used to predict local stimulus properties leading to the phenomenon of filling-in. The predicted sensory input would then be compared to the incoming sensory input, and an error signal representing the mismatch would be returned. In the absence of veridical information, no deviation and thus no error signal would occur. Effectively, the filled-in signal might have less noise. Reduced noise, in turn, results in a smaller prediction error and higher credibility at later stages. A faster reaction time to the filled-in stimulus compared to the veridical stimulus could suggest that the integration process is indeed biased with less noise. In summary, although the results reported here seem compatible with the predictive coding framework, this explanation presently remains vague and speculative.

In conclusion, we find a new behavioral effect where subjects prefer a partially inferred stimulus to a veridical one. Though both appear to be continuous, the filled-in one could hide an inset and is, therefore, less reliable. In this perceptual decision-making task, subjects do not make use of high-level assessments about the reliability of the filled-in stimulus. Even more so, they prefer the unreliable percept.

## Materials and methods

Many of the methods are taken from *Ehinger et al., 2015*. All data and analyses are available at https://osf.io/wphbd.

### Subjects

Overall, 175 subjects took part in the experiments. Of the subjects, 32% (n = 56) were removed due to the screening experiments described below. An additional 3% (n = 6) were removed due to low performance (n = 2, <75% in at least two conditions with a visible unique inset) or because they responded to the stimuli with the inset stimulus instead of the continuous stimulus (n = 4). The experimental data were not recorded in 7% (n = 13) due to eye tracking calibration problems (n = 4)

and other issues during data collection (n = 9). The remaining 100 subjects were recorded and analyzed in the following experiments.

For the first experiment, 24 subjects entered the analysis (average age 21.9 years, age range 18–28 years, 12 female, 20 right-handed, 16 right-eye dominant). Fifteen of these subjects participated in the EEG study reported by *Ehinger et al., 2015*. In the second experiment, 27 subjects entered the analysis (average age 22.4 years, age range 19–33 years, 15 female, 25 right-handed, 19 right-eye dominant). In the third, 24 subjects entered the analysis (average age 21.9 years, range 19–27 years, 19 female, 23 right-handed, 16 right-eye dominant). In the fourth experiment, we report the results of 25 subjects (average age 22.1, range 18–35, 20 female, 24 right-handed, 14 right-eye dominant). In the last experiment, the same set of subjects participated as in experiment four with the exception of a single subject, who did not finish the both parts of the combined session with experiment 4 and 5.

All subjects gave written informed consent, and the experiment was approved by the ethics committee of the Osnabrück University. For the initial experiment, we set out to analyze 25 subjects. For the second experiment, we calculated a sample size of 18 subjects based on the results of experiment one in order to have a power of 90% (calculated with gPower, (*Faul et al., 2009*), matched pair means cohen's-d = 0.72, planned-power 90%). We disclose that the results of the initial analysis with this group were not conclusive about differences between the location inside and the location above the blind spot. Although the sample size was large enough to replicate the blind spot main effect, it was not adequate to find the difference between locations. Therefore, we decided to increase the number of subjects by 50% (n = 9). For the third, fourth and fifth experiments, we used an empirical power analysis based on MLE of a linear mixed model in order to achieve 90% power for the smallest effect observed outside the blind spot. This resulted in a sample of 24 subjects.

## Screening

As described above, many subjects failed a simple screening test. In this pre-experiment, we showed a single stimulus in the periphery either inside or outside the blind spot in the left or right visual field. In two blocks of 48 trials, subjects indicated which stimulus (no inset vs. inset) had been perceived. We thought of this simple experiment to evaluate our blind spot calibration method, as an inset stimulus inside the blind spot should have been reported as no inset. The first block was used as a training block. In the second block, we evaluated the performance in a conservative way. No feedback was given to the subjects. If the performance was below 95% (three errors or more), we aborted the session because the participant was deemed to be too unreliable to proceed further with our experiment. We analyzed the data of those that failed the screening experiment, in four categories of failures that demonstrate the heterogeneity of subjects: Subjects reported inset when an inset was shown in the left blind spot (44%), or in the right blind spot (78%). Subjects did not report the inset of a stimulus presented outside the blind spot (37%), and subjects reported an inset, even though a continuous stimulus was shown (80%). The percentage represents how many subjects had at least one trial where a classification-criterion was fulfilled and thus do not add to 100%. The rates for subjects that did not fail the criterion were 16%, 21%, 13% and 22% respectively. The high percentage in the last category of removed subjects, in which they report an inset even though no inset was visible, strongly suggests that subjects failed the task not due to blind spot related issues, but due to inattention or perceptual problems. Even though we observe more wrong reports in the right than the left blind spot position, there was nevertheless no correlation with calibration position or size. Overall, 57% (n = 100) of the recruited subjects passed this test and were admitted to subsequent experiments.

## Eye tracking, screen, shutter glasses

A remote, infrared eye-tracking device (Eyelink 1000, SR Research) with a 500 Hz sampling rate was used. The average calibration error was kept below 0.5° with a maximal calibration error of 1.0°. Trials with a fixation deviation of 2.6° from the fixation point were aborted. We used a 24-inch, 120 Hz monitor (XL2420t, BenQ) with a resolution of 1920 × 1080 pixels in combination with consumer-grade shutter glasses for monocular stimulus presentation (3D Vision, Nvidia, wired version). The shutter glasses were evaluated for appropriate crosstalk/ghosting using a custom-manufactured

luminance sensor sampling at 20 kHz. The measured crosstalk at full luminance was 3.94%. The subject screen distance was 60 cm in experiment 1, 2, 4, and 5 and 50 cm in the third experiment.

## Stimuli

Modified Gabor patches with a frequency of 0.89 cycles/° and a diameter of 9.6° were generated. Two kinds of patterns were used (*Figure 1a*): one completely continuous and one with a small perpendicular inset of 2.4°. For comparison, the blind spot typically has a diameter of 4°–5°. The Gabor had constant contrast in a radius of 6.3° around the center. This ensured the same perception of the continuous stimulus outside the blind spot in comparison to a filled-in stimulus, where the inner part is inside the blind spot. To account for possible adaptation effects, horizontal and vertical stimuli were used in a balanced and randomized way across the trials. Stimuli were displayed using the Psychophysics Toolbox (*Brainard, 1997*, RRID: SCR:002881) and Eyelink Toolbox (*Cornelissen et al., 2002*). The stimuli were displayed centered at the individually calibrated blind spot location. The stimulus at the location above the blind spot in experiment two was at the same distance as the blind spot but was rotated by 25° to the horizon around the fixation cross. For the inward and outward condition of experiment 3, stimuli were moved nasally or temporally by 8.6°. Thus the stimuli had an overlap of only 1°. Less overlap is not possible without either cutting the border of the screen or overlapping with the fixation cross.

## Task

After a fixation period of 500 ms, we presented two stimuli simultaneously to the left and right of the fixation cross. Subjects were instructed to indicate via button press (left or right) which stimulus was continuous. Each stimulus was presented either in the temporal or nasal field of view. In some trials, the required response was unambiguous, when one of the stimuli showed an inset and the other did not (and the inset stimulus was presented outside the blind spot). In many trials (80% of all experiments and locations, 46% when the stimulus was shown above the blind spot in experiment 2), both stimuli were continuous and no uniquely correct answer existed (see *Figure 1—figure supplement 1* for a detailed overview of the balancing). All trials were presented in a randomized order. If the subject had not given an answer after 10 s, the trial was discarded, and the next trial started. All in all, subjects answered 720 trials over six blocks; in experiment one the trials were split up into two sessions. After each block, the eye tracker and the blind spot were re-calibrated. After cleaning trials for fixation deviation and blinks, an average of 662 trials (90%-quantile: 585, 710) remained. For two subjects, only 360 trials could be recorded.

## Bootstrap in figures

In several figures, we present data with summary statistics. To construct the confidence intervals we used bias-corrected, accelerated 95% bootstrapped confidence intervals of the mean with 10,000 resamples. Note that the summary statistics do not need to conform to the posterior summary estimates because they are marginals. Only the posterior model values reflect the estimated effect.

## Blind spots

In order to calibrate the blind spot locations, subjects were instructed to use the keyboard to move a circular monocular probe on the monitor and to adjust its size and location to fill the blind spot with the maximal size. They were explicitly instructed to calibrate it as small as necessary to preclude any residual flickering. The circular probe flickered from dark gray to light gray to be more salient than a probe with constant color (*Awater et al., 2005*). All stimuli were presented centered at the respective calibrated blind spot location. In total, each subject calibrated the blind spot six times. For the following comparisons of blind spot characteristics, we evaluated one-sample tests with the percentile bootstrap method (10,000 resamples) of trimmed means (20%) with alpha = 0.05 (*Wilcox, 2012*). For paired two-sample data, we used the same procedure on the difference scores and bias-corrected, accelerated 95% bootstrapped confidence intervals of the trimmed mean (20%). We report all data combined over all experiments. In line with previous studies (*Wolf and Morandi, 1962*; *Ehinger et al., 2015*), the left and right blind spots were located horizontally at −15.52° (SD = 0.57° CI:[−15.69°,−15.36°]) and 15.88° (SD = 0.61° CI:[15.70°,16.07°]) from the fixation cross.

The mean calibrated diameter was 4.82° (SD = 0.45° CI:[4.69°,4.95°]) for the left and 4.93° (SD = 0.46° CI:[4.79°,5.07°]) for the right blind spot. Left and right blind spots did significantly differ in size (p=0.009, CI:[−0.17°,−0.03°] and in absolute horizontal position (in relation to the fixation cross; p<0.001, CI: [0.27°,0.45°]). On average, the right blind spot was 0.36° further outside of the fixation cross. No significant difference was found in the vertical direction (p=0.37), but this is likely due to the oval shape of the blind spot in this dimension and the usage of a circle to probe the blind spot. These effects seem small, did not affect the purpose of the experiments and will not be discussed further.

## GLMM analysis

We fitted a Bayesian logistic mixed effects model predicting the probability of responding 'right' with multiple factors that represent the temporal over nasal bias and several other covariates described below. Because we were interested in the bias between the nasal fields and the temporal fields of view, we combined both predictors for the left and right temporal (and nasal, respectively) locations and reported the combined value.

Data were analyzed using a hierarchical logistic mixed effects models fitted by the No-U-Turn Sampler (NUTS, STAN Development Team). The model specification was based on an implementation by Sorensen, Hohenstein and Vasisth (*Sorensen et al., 2016*). In the results section, we report estimates of linear models with the appropriate parameters fitted on data of each experiment independently. We also analyzed all data in one combined model: there were no substantial differences between the results from the combined model and the respective submodels (*Supplementary file 1*). The models are defined as follows using the Wilkinson notation:

$$answer_{Right} \sim 1 + Temporal_{Left} * Location + Temporal_{Right} * Location + Answer_{Right}(t-1) +$$
$$Handedness_{Right} + DominantEye_{Right} +$$
$$(1 + Temporal_{Left} * Location + Temporal_{Right} * Location + Answer_{Right}(t-1) | Subject)$$
$$Answer_{i_{Right}} \sim Bernoulli(\theta_i)$$
$$\theta_i = logit^{-1}(X_{within}\beta_{within} + X_{between}\beta_{between} + N(0, \tau X_{within}) + N(0, e_i))$$

Two factors were between subjects: *handedness* and *dominant eye*. In total, we have four within-subject factors, resulting in eight parameters: There are two main factors representing whether the left, and respectively the right, stimulus was inside or outside the *temporal* field. Depending on the experiment, the main factor *location* had up to three levels: the stimuli were presented outward (Exp. 3), inward (Exp. 3), above (Exp 2, 5) or on (all experiments) the blind spot. In addition, we modeled the interactions between location and whether the left stimulus (and the right stimulus, respectively) was shown temporally. In order to assure independence of observation, an additional within-subject main factor *answer(t-1)* was introduced, which models the current answer based on the previous one. In frequentist linear modeling terms, all within-subject effects were modeled using random slopes clustered by subject and a random intercept for the subjects. We used treatment coding for all factors and interpreted the coefficients accordingly.

In the model, we estimated the left and right temporal field effects separately. For the statistical analysis, we combined these estimates by inverting the left temporal effect and averaging with the right temporal effect. We did this for all samples of the mcmc-chain and took the median value. We then transformed these values to the probability domain using the inverse-logit function, subtracting the values from 0.5 and multiplying by 100. All results were still in the linear range of the logit function. We calculated 95% credible intervals the same way and reported them as parameter estimates ($CDI_{95}$ lower-upper) in the text. These transformed values represent the additive probability (in %) of choosing a left (right) stimulus that is shown in the left (right) temporal field of view compared to presenting the left (right) stimulus in the nasal field of view, keeping all other factors constant.

## Reaction times

Initially, we did not plan to analyze the reaction time data. These analyses are purely explorative. The response setup consisted of a consumer keyboard. Thus delays and jitters are to be expected. However, with an average of 494 ambiguous trials per subject, we did not expect a spurious bias between conditions due to a potential jitter. Reaction time data was analyzed with a simple Bayesian mixed linear model:

$$RT \sim 1 + Temporal_{selected} * Location + (1 + Temporal_{selected} * Location | Subject)$$

Only trials without a visible inset stimulus were used. *Temporal selected* consists of all trials where a temporal stimulus was selected. Because of the bias described in the results, there was no imbalance between the number of trials in the two conditions (difference of 10 trials bootstrapped-CI [−2, 23]).

## Bayesian fit

We did not make use of prior information in the analysis of our data. We placed implicit, improper, uniform priors from negative to positive infinity on the mean and 0 to infinity for the standard deviations of our parameters, the default priors of STAN. An uninformative lkj-prior ($\nu = 2$) was used for the correlation matrix, slightly emphasizing the diagonal over the off-diagonal of the correlation matrix (*Sorensen et al., 2016*; *Carpenter et al., 2017*).

We used six mcmc-chains using 2000 iterations each, with 50% used for the warm-up period. We visually confirmed convergence through autocorrelation functions and trace plots, then calculated the scale reduction factors (*Gelman et al., 2014*), which indicated convergence (Rhat < 1.1).

## Posterior predictive model checking

Posterior predictive model checks were evaluated to test for model adequacy (*Gelman et al., 2013*). Posterior predictive checks work on the rationale that newly generated data based on the model fit should be indistinguishable from the data that the model was fitted by originally. Due to our hierarchical mixed model, we perform posterior predictive checks on two levels: trial, and subject. In the first case, we generate new datasets (100 samples) based on the posterior estimates of each subject's effect. We compare the distribution of this predicted data with the actual observed values for each (*Figure 6—figure supplement 2a*). At the subject level, we draw completely new data sets, based on the multivariate normal distribution given by the random effects structure. We then compare the collapsed blind spot effect once for the newly drawn subjects with the observed data (*Figure 6—figure supplement 2b*). Taken together, these posterior predictive model checks show that we adequately capture the very diverse behavior of our subjects but also correctly model the blind spot effect on a population basis.

## Effects not reported in the result section

Here we report the result of the covariate factor based on the combined model (all experiments modeled together). Note that the interpretation of such effects naturally occurs on logit-transformed values. Summation of different parameter-levels (as necessary for treatment coding) on logit-scale can be very different to summations of raw-percentage values. It can also be similar, close to the linear scale of the logit-transform, that is, close to 50% (which we made use of for the blind spot effects reported at other points of the manuscript). We did not find evidence for a different global bias (main effect location) in any of the four stimulation positions tested here. The dominant eye factor had an 11.51% effect ($CDI_{95}$2.78–19.59%) on the global bias. Thus subjects with a dominant right eye also showed a preference to the right stimulus over the left one, irrespective of whether the stimulus was visible through the left or the right eye. We find a global bias (the intercept, −26.75% $CDI_{95}$−38.18% to −9.29%, with treatment coding) toward choosing the left stimulus; this might reflect that in the first two experiments we instructed subjects to use the right hand, thus they used their index and middle fingers. In the third experiment we instructed subjects to use both index fingers, resulting in a decreased bias to the left, with a shift more to the right, and thus more to balanced answers, of 12.24% ($CDI_{95}$−1.98–24.16%]). We did not find evidence for a bias due to handedness (7.71%, $CDI_{95}$−8.96–22.75%). There was an influence of the previous answer on the current answer. We observe a global effect of 7.86% ($CDI_{95}$ 0.53%–14.95%), which suggests that subjects are more likely to choose e.g. the right stimulus again when they have just chosen 'right' in the previous trial. For this effect it is more important to look at random effect variance, which is quite high with a standard deviation of −31.4 ($CDI_{95}$28.27–34.69%), suggesting that there is a large variation between subjects. Indeed, a closer look at the random slopes of the effect reveals three different strategies: Some subjects tend to stick the same answer, some subjects are balanced in their answers without any trend, and some subjects tend to regularly alternate their answers in each trial.

Note that this behavior does not seem to influence any of the other effects: We do not see any correlation between the random effects, except for the correlation between the n-1 effect and the intercept ($-0.55$, CI: $-0.72,-0.34$). This correlation means that subjects who tend to alternate their key presses will not have a strong bias in the intercept, or the other way around, subjects who press the same key all the time also have a bias towards this key.

Other extended models we considered showed no effect when both stimuli were in the temporal field, nor any three-way interaction. Following standard procedures to avoid spurious effects of unnecessary degrees of freedom, we removed these variables from the final model.

## Acknowledgements

We thank Tim C Kietzmann for his valuable suggestions on the design of the experiment and Frank Jäkel for his valuable discussions of the analysis. This work was supported by SPP 1665 (BE, PK), the European Union through the H2020 FET Proactive project socSMCs (GA no 641321, BE, PK) and SFB 936, project B1 (JO).

## Additional information

### Funding

| Funder | Grant reference number | Author |
| --- | --- | --- |
| Horizon 2020 | socSMC 641321 | Benedikt V Ehinger<br>Peter König |
| Deutsche Forschungsgemeinschaft | SPP 1665 | Benedikt V Ehinger<br>Peter König |
| Deutsche Forschungsgemeinschaft | SFB 936 B1 | José P Ossandón |

The funders had no role in study design, data collection and interpretation, or the decision to submit the work for publication.

### Author contributions

BVE, Conceptualization, Data curation, Software, Formal analysis, Validation, Investigation, Visualization, Methodology, Writing—original draft, Project administration, Writing—review and editing; KH, Data curation, Software, Formal analysis, Investigation, Visualization, Methodology; JPO, Conceptualization, Supervision, Validation, Investigation, Visualization, Methodology, Writing—original draft, Project administration, Writing—review and editing; PK, Conceptualization, Supervision, Funding acquisition, Validation, Writing—original draft, Project administration, Writing—review and editing

### Author ORCIDs

Benedikt V Ehinger, http://orcid.org/0000-0002-6276-3332
José P Ossandón, http://orcid.org/0000-0002-2539-390X
Peter König, http://orcid.org/0000-0003-3654-5267

### Ethics

Human subjects: All subjects gave written informed consent, and the experiment was approved by the ethics committee of the University of Osnabrück.

## Additional files

### Supplementary files

• Supplementary file 1. Overview of the results of all experiments individually and the combined estimates. Empty cells indicate that the condition was not measured in this study.

## Major datasets

The following dataset was generated:

| Author(s) | Year | Dataset title | Dataset URL | Database, license, and accessibility information |
|---|---|---|---|---|
| Benedikt V Ehinger, Katja Häusser, José P Ossandón, Peter König | 2017 | A Perceptual Bias in the Blind SpotPublic | https://osf.io/wphbd/ | CC-By Attribution 4.0 International |

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
