## [Decision Letter]

Thank you for submitting your article "Mind over matter: A perceptual decision bias toward filled-in stimuli in the blind spot" for consideration by *eLife*. Your article has been reviewed by three peer reviewers, and the evaluation has been overseen by a Reviewing Editor and David Van Essen as the Senior Editor. The following individual involved in review of your submission has agreed to reveal his identity: Jeff Beck (Reviewer #1).

The reviewers have discussed the reviews with one another and the Reviewing Editor has drafted this decision to help you prepare a revised submission.

Summary:

This manuscript investigates whether there is bias towards a target with more complete sensory input compared to a less informative stimulus that is "filled-in". The major finding is that the blindspot-covering stimulus (regardless of whether it included an orthogonal inset) was selected in favor of the completely viewed stimulus. This is an unexpected and intriguing finding, and seems like *eLife* material. However, the reviewers had a number of comments, many related to clarity. In addition, it seems that one more control experiment is needed to really nail this result.

Essential revisions:

1) In the first experiment, ten trial types were used (shown in Figure 2). What varied was whether either of the stimuli included an inset, and whether they were presented in the blindspot. It is not clear, however, how many times each of these trial types were used. This may be relevant because if pooling across trials reveals a bias such that presentation of a stimulus inside the blindspot is actually predictive of it being the correct choice, then of course it is not suboptimal for subjects to be biased towards the blindspot stimulus.

2) During screening, when subjects were shown "a single stimulus in the periphery either inside or outside the blindspot in the left or right visual field", the first block of trials was used as training. The description of this procedure ("an inset stimulus inside the blind spot should have been reported as no inset") suggests that subjects received corrective feedback, such that every time a blindspot stimulus was shown the correct response was "no inset" – regardless of whether there was or wasn't an inset in the actual stimulus shown. If this was the case, then this training could explain the bias of subjects preferring the blindspot stimulus in later phases of the experiment(s). Please explain and discuss how this potential confound can be excluded based on your current results, or run a new experiment in which this is controlled for.

3) Finally, one control was notably missing from the experiments: a variant in which subjects have to choose the stimulus *with* (rather than without) the inset. If the authors' conclusions are correct, the observed biases should be reversed: the blindspot stimulus should be chosen *less* than the non-blindspot stimulus in ambiguous trials. Experiments should be performed to confirm this. We do not take lightly the requirement of additional experiments, but in this case the result is so unexpected (a good thing) that it's necessary to rule out all reasonably likely explanations.

4) It seems like the main analysis for the accuracy results was a linear mixed model fit. However, the only time this is mentioned in the main text is the caption of Figure 3. Mentioning it so briefly is misleading, since that makes it seem like the linear model was used only for this later analysis (and another for the reaction time analyses). In addition, earlier captions refer to a bootstrap method suggesting that method was the main analysis. Secondarily, the bootstrap method is not clearly distinguished in the Methods. It is possibly buried in the section about blindspot calibration.

We'll leave it up the authors how to clarify this. One possibility is to introduce the linear model early in the Results, since this seems essential to interpreting the Results. The other is to put a clear explanation in Methods, and refer to it in results where appropriate. In any case, the bootstrap method should be clearly distinguished and explained (avoiding jargon) in the Methods.

5) There are a number of covariates that the authors seem to include in the linear model. These are buried in the other findings in the Methods (subsection “Effects not reported in the result section”). For example, there are eye dominance, perseverative (more likely to repeat an answer), or stereotypical (alternate answers, stick to the same answer) effects. These should be clearly explained, all in one place, either in the Results or Methods (if it's the latter, with appropriate pointers in the main text). The authors should also think about presenting the screening test results in the main text, including evidence (as alluded to in the subsection “Screening”) that failed screening was not related to the blindspot calibration.

6) Some goodness of fit measure should be given for the model fits.

7) The authors should provide some significance testing, including details on how these are calculated.

8) It seems that during a single test that only one eye is receiving input, while the other is completely blinded (not receiving visual input). Please make this more explicit in the text. Instead of using jargon like monocular display in the second paragraph of the Results. Explicitly state that for each test one eye sees both targets while the other is completely blinded and then point to Figure 1. If both eyes are receiving input, as possibly suggested in the seventh paragraph of the Results (mention of binocular rivalry) then this is completely unclear and needs to be clarified earlier in the text and Figure 1.

9) There is an alternative explanation of these results that is somewhat orthogonal to the predictive coding explanation. Throughout experiment 1 subjects are shown patches at two different locations and those patches occur with equal frequency, but are never perceived to occur in the blindspot. Nonetheless, subjects maintain a strong prior belief that the target should occur with equal frequency at each location and so in the presence of enough ambiguity they may be inclined to 'guess' in the place where they have never seen the target, either because of the gambler's fallacy, or because of probability matching. I think the math actually checks out. There are 10 experimental conditions 2 of which are unambiguous providing 1 confirmed vote that the target has occurred outside left and one for outside right. If you want to end up guessing that the target as occurred at each location inside/outside/left/right you need a spurious guess at the blindspot location when the alternative was a non-blindspot location which would predict a behavioral bias of 100% toward the blindspot in that condition. Therefore, any inclination toward probability matching or a gamblers fallacy would have predicted this result.

We admit that this is a bit speculative, but it's something the authors might want to think about. To look for probability matching and gamblers fallacy, one need only to look at the effects of the previous trials on the current one. If a subject has chosen A a few more times than average this should create a bias toward B. If that's not the case, then this explanation can be ruled out.

10) The Discussion (and the corresponding part of the Abstract) linking these results to predictive coding was confusing – even in the best case, pretty substantial additional assumptions are needed to accommodate these results within the framework of predictive coding. It might be better to publish these results on face value, as they are, rather than trying to squeeze them into the procrustean framework of an inappropriate theory. The contrast between predictive coding and theories positing "a hierarchy of increasingly refined representations", which is only referred to in the Abstract as if this study was specifically testing it – which is obviously not the case – is particularly unclear.

---

## [Author Response]

Essential revisions:

1) In the first experiment, ten trial types were used (shown in Figure 2). What varied was whether either of the stimuli included an inset, and whether they were presented in the blindspot. It is not clear, however, how many times each of these trial types were used. This may be relevant because if pooling across trials reveals a bias such that presentation of a stimulus inside the blindspot is actually predictive of it being the correct choice, then of course it is not suboptimal for subjects to be biased towards the blindspot stimulus.

Done. We provide a new supplementary figure that details the randomization of all experiments. This includes the three experiments presented in the initial manuscript plus two new controls. Now it is possible to evaluate in detail that the described effects are not produced by a bias in the trial pooling (see the answer to comment 9 for further explanations). We clarified the manuscript and added the corresponding links to the randomization figure where appropriate. The complexity of the trial pooling is now better communicated (see subsections “Experiment 5”, “Task” and the new Figure 1—figure supplement 1).

2) During screening, when subjects were shown "a single stimulus in the periphery either inside or outside the blindspot in the left or right visual field", the first block of trials was used as training. The description of this procedure ("an inset stimulus inside the blind spot should have been reported as no inset") suggests that subjects received corrective feedback, such that every time a blindspot stimulus was shown the correct response was "no inset" – regardless of whether there was or wasn't an inset in the actual stimulus shown. If this was the case, then this training could explain the bias of subjects preferring the blindspot stimulus in later phases of the experiment(s). Please explain and discuss how this potential confound can be excluded based on your current results, or run a new experiment in which this is controlled for.

Done. We apologize for this misunderstanding. Subjects did not receive any feedback. This is now explicit in the manuscript. We also report more detailed on the reasons why subjects failed our screen experiment. See subsection “Screening” and the answer to 5.

3) Finally, one control was notably missing from the experiments: a variant in which subjects have to choose the stimulus with (rather than without) the inset. If the authors' conclusions are correct, the observed biases should be reversed: the blindspot stimulus should be chosen less than the non-blindspot stimulus in ambiguous trials. Experiments should be performed to confirm this. We do not take lightly the requirement of additional experiments, but in this case the result is so unexpected (a good thing) that it's necessary to rule out all reasonably likely explanations.

Done. We agree that this additional control is useful and performed as requested. The control experiment confirms our prediction. This new control (Experiment 4) is an exact repetition of experiment 1, except that subjects were requested to select the stimulus with the inset instead of the continuous one. As predicted, the bias was reversed; in ambiguous diagnostic trials, subjects showed a bias for the non-blind spot stimulus. This result indicates that subjects were indeed trying to follow the instructions during ambiguous trials, and strengthen our conclusion that, in the original experiments, they chose the blind spot stimulus because it is considered a better exemplar of a continuous stimulus. This result also excludes explanations based on non-specific biases for the blind spot stimulus. This is now described in the Results subsection (“Experiment 4”) in Figure 4 and in the Discussion (third paragraph). Additionally, subjects performing experiment 4 were identical to the subjects performing experiment 5 (Experiment 5 was always performed after experiment 4). In the later experiment, they performed the original task, choose the stimulus without inset. Thus, we have data for both tasks within the same subjects. We observe a strong negative correlation. Thus, indication that subjects who biased their choices towards the stimulus outside of the blind spot in experiment 4, biased their choices towards the stimulus inside the blind spot in experiment 5 (as found before in Experiment 1-3).

4) It seems like the main analysis for the accuracy results was a linear mixed model fit. However, the only time this is mentioned in the main text is the caption of Figure 3. Mentioning it so briefly is misleading, since that makes it seem like the linear model was used only for this later analysis (and another for the reaction time analyses). In addition, earlier captions refer to a bootstrap method suggesting that method was the main analysis. Secondarily, the bootstrap method is not clearly distinguished in the Methods. It is possibly buried in the section about blindspot calibration.

We'll leave it up the authors how to clarify this. One possibility is to introduce the linear model early in the Results, since this seems essential to interpreting the Results. The other is to put a clear explanation in Methods, and refer to it in results where appropriate. In any case, the bootstrap method should be clearly distinguished and explained (avoiding jargon) in the Methods.

Done. There was a confusion when we used the linear mixed model and when we used a bootstrapping procedure. We clarified in the manuscript that all our conclusions are only based on the Generalized Linear Mixed Model Line (subsection “Experiment 1”, fourth paragraph). We used the bootstrapping procedure only to visualize summary statistics in the figures and to summarize the location and sizes of the blind spots as described in the respective Methods section. We improved the manuscript and made it clear in the captions, which values are bootstrapped marginal summaries and what values represent posterior model estimates. The bootstrap used for visual display is now described more prominently in the Methods section (subsection “Bootstrap in figures”). In addition, we introduce the linear model early in the Results part (subsection “Experiment 1”, fourth paragraph).

5) There are a number of covariates that the authors seem to include in the linear model. These are buried in the other findings in the Methods (subsection “Effects not reported in the result section”). For example, there are eye dominance, perseverative (more likely to repeat an answer), or stereotypical (alternate answers, stick to the same answer) effects. These should be clearly explained, all in one place, either in the Results or Methods (if it's the latter, with appropriate pointers in the main text).

Done. Thank you for this excellent suggestion. We added clear pointers in the manuscript (subsection “Experiment 1”, fourth paragraph). We also improved upon the description part in the Methods (subsection “Effects not reported in the result section”).

The authors should also think about presenting the screening test results in the main text, including evidence (as alluded to in the subsection “Screening”) that failed screening was not related to the blindspot calibration.

Done. We classified the failed screening subjects in multiple categories. See subsection “Screening”.

6) Some goodness of fit measure should be given for the model fits.

Done. We added a new supplementary figure illustrating the multiple different posterior predictive checks we performed. Our model captures the main portions of the data well both at the subject and group levels. We also added a paragraph the text that clearly describes our procedure and rationale for the posterior predictive checks (subsection “Posterior predictive model checking” and Figure 6—figure supplement 2).

7) The authors should provide some significance testing, including details on how these are calculated.

Done. While a Bayesian quantification of the posterior can be used for inference on the data, we additionally performed t-tests on the main two questions: Is there a bias for the blind spot (p<0.001) and is it greater than in the other tested locations (p<0.001). These tests can be found in subsection “Combined effect estimates over all experiments”.

8) It seems that during a single test that only one eye is receiving input, while the other is completely blinded (not receiving visual input). Please make this more explicit in the text. Instead of using jargon like monocular display in the second paragraph of the Results. Explicitly state that for each test one eye sees both targets while the other is completely blinded and then point to Figure 1. If both eyes are receiving input, as possibly suggested in the seventh paragraph of the Results (mention of binocular rivalry) then this is completely unclear and needs to be clarified earlier in the text and Figure 1.

Done. This was indeed worded a bit confusing. We improved on the passages mentioned: If one stimulus is partially inside the blind spot and the other outside, they are always shown in one eye. In other conditions (e.g. both stimuli outside the blind spot) stimuli could have been presented been presented to both eyes (but in this case, each stimulus to only one eye). We improved the text (subsection “Experiment 1”, first paragraph) and included a new Panel in Figure 1.

9) There is an alternative explanation of these results that is somewhat orthogonal to the predictive coding explanation. Throughout experiment 1 subjects are shown patches at two different locations and those patches occur with equal frequency, but are never perceived to occur in the blindspot. Nonetheless, subjects maintain a strong prior belief that the target should occur with equal frequency at each location and so in the presence of enough ambiguity they may be inclined to 'guess' in the place where they have never seen the target, either because of the gambler's fallacy, or because of probability matching. I think the math actually checks out. There are 10 experimental conditions 2 of which are unambiguous providing 1 confirmed vote that the target has occurred outside left and one for outside right. If you want to end up guessing that the target as occurred at each location inside/outside/left/right you need a spurious guess at the blindspot location when the alternative was a non-blindspot location which would predict a behavioral bias of 100% toward the blindspot in that condition. Therefore, any inclination toward probability matching or a gamblers fallacy would have predicted this result.

We admit that this is a bit speculative, but it's something the authors might want to think about. To look for probability matching and gamblers fallacy, one need only to look at the effects of the previous trials on the current one. If a subject has chosen A a few more times than average this should create a bias toward B. If that's not the case, then this explanation can be ruled out.

Done. This comment raised a possible confound that we did not consider before and that potentially could invalidate the purpose and interpretation of the experiment (i.e., that the effects are not related to the blind spot and filling-in phenomena but only to the probability structure of the task). Thus, we are thankful for the given opportunity to evaluate, and finally rule-out, these alternative explanations. We do this by an analysis of data presented before as well as a new experiment. As the gambler fallacy and probability matching explanation predict different behaviors we addressed them separately.

The results cannot be explained by the gambler fallacy. This type of bias is for the non-experienced alternative, so, as during unambiguous trials subjects never experienced an inset in the blind spot location, their ‘gamble’ in ambiguous diagnostic trials (one stimulus inside the blind spot and one outside) would have to be to choose the stimulus outside the blind spot. As the result shows the opposite bias, we can safely disregard this option.

A probability matching explanation had a much stronger case as an alternative explanation for the results presented here. As correctly noted by the reviewers, subjects never experience an inset in the blind spot during unambiguous trials, thus representing a base rate of 100% non-inset stimulus in the blind spot. Then, in ambiguous trials subjects might respond based on this base rate resulting in a bias in the same direction as observed. However, we can exclude this explanation based on a detailed analysis of the trial pooling of the control experiment 2 and on the results of a new control (experiment 5).

The trial pooling of the control positions in experiment 2 holds some evidence against a probability matching explanation. If the subjects were biased according to the base rate of continuous stimulus during the unambiguous trials, this should also be evident in the control position (‘above’). There are two different ways a base rate can be constructed that would nevertheless result in similar behaviors. In the first case, the base rate is calculated individually for each position in term of the probability of displaying a continuous stimulus during unambiguous trials. In experiment 2, the base rate for continuous stimulus in the above position, is 0.41 (84/120, Figure 1—figure supplement 1, 2^nd^ row, columns 6-8 and 10-12) for temporal stimuli and.6 (108/180, Figure 1—figure supplement 1, 2^nd^ row, columns 5-7 and 9-11) for nasal stimulus (these are the probabilities of a stimulus in a given position being continuous and do not need to add up, since there are trials in which both stimuli are in the temporal or nasal fields). Accordingly, during ambiguous trials, this will predict a slight bias for the nasal position in the control above position, which is opposite to what was observed. In the second case, the base rate is calculated for the relevant position contrasts (the diagnostic trials in which stimuli are presented to the temporal and nasal field of the same eye). In this case, the base rate during unambiguous trial is.33 (36/108, Figure 1—figure supplement 1, 2^nd^ row, columns 6-7 and 10-11) for a continuous stimulus being presented on the temporal location and the complement.77 (72/108, same) for the nasal position, thus again predicting a bias for the nasal location in opposition to what we observed. In summary, the behavior observed for the control locations indicated that subjects were not using a probability matching strategy during ambiguous trials.

Although the above analysis already indicates that a probability matching explanation is unlikely, it would require considerable effort by the readers to evaluate it, and we did not feel completely satisfied by it. Therefore, to ensure that probability matching was not driving the observed bias we performed a new control experiment. This was a repetition of experiment 2 but with a different pooling structure. The crucial difference was that in the new control, subjects never experience an inset in the temporal field control positions ‘above’ (red frame columns 6,8,11, and 12 in Figure 1—figure supplement 1). This makes the probability structure of the control position trials equivalent to one of the blind spot trials. Therefore, if the bias observed was due to probability matching, this should now be equal in the control position. The result of this control experiment shows that (1) we replicate for the fourth time the bias to the blind spot (with a new group of subjects), (2) the control position show a reduced bias compared to the blind spot (same result as in experiment 2), thus discarding that the effect in the blind spot is due to a probability matching bias.

The rationale to exclude a probability matching explanation and the new control is now described in the Results subsection (“Experiment 5”), in Figure 4, and Figure 4—figure supplement 1.

10) The Discussion (and the corresponding part of the Abstract) linking these results to predictive coding was confusing – even in the best case, pretty substantial additional assumptions are needed to accommodate these results within the framework of predictive coding. It might be better to publish these results on face value, as they are, rather than trying to squeeze them into the procrustean framework of an inappropriate theory. The contrast between predictive coding and theories positing "a hierarchy of increasingly refined representations", which is only referred to in the Abstract as if this study was specifically testing it – which is obviously not the case – is particularly unclear.

Done. We agree that the previous presentation was too strong in terms of testing the predictive coding framework (although we view it slightly more benevolent) and we revised them accordingly. Specifically, we simplified the Discussion and focused it on why the results are unexpected. It provides an explicitly speculative explanation on why subjects choose suboptimally but we removed all theoretical links that were before not well substantiated. We still think that predictive coding offers a very useful way to think about this problem and discuss it appropriately (Discussion, fourth paragraph). The Abstract and Introduction were changed correspondingly.